



# Geostatistical inverse modeling with very large datasets: an example from the OCO-2 satellite

Scot M. Miller[1], Arvind K. Saibaba[2], Michael E. Trudeau[3], Marikate E. Mountain[4], and Arlyn E. Andrews[3]

[1]Department of Environmental Health and Engineering, Johns Hopkins University, Baltimore, MD, USA
[2]Department of Mathematics, North Carolina State University, Raleigh, NC, USA
[3]Global Monitoring Division, National Oceanic and Atmospheric Administration, Boulder, CO, USA
[4]Atmospheric and Environmental Research, Inc., Lexington, MA, USA

**Correspondence:** Scot M. Miller (smill191@jhu.edu, scot.m.miller@gmail.com)

**Abstract.** Geostatistical inverse modeling (GIM) has become a common approach to estimating greenhouse gas fluxes at the Earth's surface using atmospheric observations. GIMs are unique relative to other commonly-used approaches because they do not require a single emissions inventory or a bottom-up model to serve as an initial guess of the fluxes. Instead, a modeler can incorporate a wide range of environmental, economic, and/or land use data to estimate the fluxes. Traditionally, GIMs have been paired with in situ observations that number in the thousands or tens of thousands. However, the number of available atmospheric greenhouse gas observations has been increasing enormously as the number of satellites, airborne measurement campaigns, and in situ monitoring stations continues to increase. This era of prolific greenhouse gas observations presents computational and statistical challenges for inverse modeling frameworks that have traditionally been paired with a limited number of in situ monitoring sites. In this article, we discuss the challenges of estimating greenhouse gas fluxes using large atmospheric datasets with a particular focus on GIMs. We subsequently discuss several strategies for estimating the fluxes and quantifying uncertainties, strategies that are adapted from hydrology, applied math, or other academic fields and are compatible with a wide variety of atmospheric models. We further evaluate the accuracy and computational burden of each strategy using $CO_2$ observations from NASA's Orbiting Carbon Observatory 2 (OCO-2) satellite. Specifically, we simultaneously estimate a full year of 3-hourly $CO_2$ fluxes across North America in one case study – a total of $9.4 \times 10^6$ unknown fluxes using $9.9 \times 10^4$ observations. The strategies discussed here provide accurate estimates of $CO_2$ fluxes that are comparable to fluxes calculated directly or analytically. We are also able to approximate posterior uncertainties in the fluxes, but these approximation are typically an over- or underestimate depending upon the strategy employed and the degree of approximation required to make the calculations manageable.

## 1 Introduction

Atmospheric observations of air pollutants and greenhouse gases have evolved dramatically over the past decade. Atmospheric monitoring of carbon dioxide ($CO_2$) is a prime example. The number of in situ observation sites in the US, Canada, Europe, and elsewhere has greatly expanded since the early 2000s. For example, a recent geostatistical inverse modeling (GIM) study





of $CO_2$ fluxes across North America used observations from six times as many continuous tower-based observation sites as a GIM study of the same region published six years earlier (Gourdji et al., 2012; Shiga et al., 2018). Aircraft-based observations have also greatly expanded, including regular observations from civilian aircraft based in both Germany and Japan (Petzold et al., 2015; Machida et al., 2008).

Several $CO_2$-observing satellites have also launched in the past decade, greatly expanding both the quantity and spatial extent of atmospheric $CO_2$ observations. For example, the Greenhouse Gases Observing Satellite (GOSAT) launched in 2009 (e.g., Butz et al., 2011), OCO-2 in 2014 (e.g., Eldering et al., 2017), TanSat in 2016 (Yang et al., 2018), GOSAT-2 in 2018 (e.g., Masakatsu Nakajima, 2012), and OCO-3 in 2019 (e.g., Eldering et al., 2019).

These new in situ and remote sensing datasets provide an unprecedented new window into surface fluxes of $CO_2$ and
other atmospheric trace gases. However, the sheer quantity and geographic scope of the data present enormous computational challenges for inverse modeling frameworks that estimate trace gas emissions. The OCO-2 satellite, for example, collects approximately $2 \times 10^6$ $CO_2$ observations per month that pass quality screening (Eldering et al., 2017). The number of total available observations will only increase as the current fleet of $CO_2$-observing satellites continue to collect observations and as additional satellites launch into orbit. Approaches to inverse modeling that are well-suited to "big data" will arguably be able
to make the most of these new datasets to constrain $CO_2$ sources and sinks.

This paper discusses the challenges of using large atmospheric air pollution and greenhouse gas datasets through the lens of GIMs. A GIM is unique relative to a classical Bayesian inversion that uses a prior emissions inventory or a bottom-up flux model. In place of a traditional prior emissions estimate, a GIM can incorporate a wide variety of environmental, economic, or population data that may help predict the distribution of surface fluxes (e.g., Gourdji et al., 2012; Miller et al., 2013, 2016;
Shiga et al., 2018). The GIM will then weight each of these predictor datasets to best match the atmospheric observations. A GIM will further estimate grid-scale flux patterns that are implied by the atmospheric observations but do not match any patterns in the predictor datasets. For example, existing GIM studies have used predictors drawn from reanalysis products and satellites, including air temperature, soil moisture, and solar-induced fluorescence (SIF) (e.g., Gourdji et al., 2012; Miller et al., 2016; Shiga et al., 2018). Other studies have used predictors of anthropogenic activity, including maps of human population
density and agricultural activity (e.g., Miller et al., 2013). In the alternative, one can also build a GIM without any predictor datasets (e.g., Michalak et al., 2004; Mueller et al., 2008; Miller et al., 2012). In this case, the GIM will rely entirely on the atmospheric observations to estimate the surface fluxes.

The purpose of this study is to adapt recent computational innovations in inverse modeling from other academic disciplines, including hydrology and seismology, to GIMs of atmospheric gases. The primary goal is to develop inverse modeling strategies
that can assimilate very large atmospheric datasets. An additional aim of this work is to develop flexible approaches that can be paired with many different types of atmospheric models (e.g., gridded, Eulerian models and particle-following models). To this end, we first provide an overview of GIMs and the specific challenges posed by large datasets. We subsequently discuss two options for calculating the best estimate of the fluxes and two options for estimating the posterior uncertainties, and we have published the associated code in a public repository (Miller and Saibaba, 2019). Lastly, we develop a case study using the





OCO-2 satellite as a lens to evaluate the advantages and drawbacks of each approach. OCO-2 collects millions of observations

per year and is therefore prototypical of many current, and likely future, big data inverse modeling problems.

## 2   Context on the geostatistical approach to inverse modeling

A GIM will estimate a set of fluxes $s$ (dimensions $m \times 1$) that match atmospheric observations $z$ ($n \times 1$), using an atmospheric

transport model $\mathbf{H}$ ($n \times m$):

$$z = \mathbf{H}s + \epsilon \qquad (1)$$

The fluxes ($s$), when passed through the atmospheric model ($\mathbf{H}$) will never exactly match the data ($z$) due to a variety of errors

($\epsilon$, dimensions $n \times 1$), including errors in the measurements ($z$) and errors in the atmospheric transport model ($\mathbf{H}$). However,

$\mathbf{H}s$ should match the observations within a specified error, and many inverse models, including a GIM, require that the modeler

input a covariance matrix defining the characteristics of these errors (e.g., Rodgers, 2000; Michalak et al., 2004):

$$\epsilon \sim \mathcal{N}(\mathbf{0}, \mathbf{R}) \qquad (2)$$

where $\sim$ means "is distributed as," $\mathcal{N}$ is a multivariate normal distribution, and $\mathbf{R}$ ($n \times n$) is the covariance matrix that must

be defined by the modeler before running the GIM.

Furthermore, the fluxes ($s$) in a GIM have two different components, and both components are estimated as part of the

inverse model (e.g., Kitanidis and Vomvoris, 1983; Michalak et al., 2004):

$$s = \mathbf{X}\boldsymbol{\beta} + \boldsymbol{\zeta} \qquad (3)$$

where $\mathbf{X}$ ($m \times p$) is a matrix of $p$ predictor datasets or covariates that may help describe patterns in the unknown fluxes ($s$)

(refer to Sect. 1). The coefficients $\boldsymbol{\beta}$ ($p \times 1$) will scale the variables in $\mathbf{X}$. These coefficients are unknown and estimated as

part of the GIM. Collectively, $\mathbf{X}\boldsymbol{\beta}$ is referred to as the model of the trend or the deterministic model. Furthermore, $\boldsymbol{\zeta}$ ($m \times 1$)

contains grid-scale patterns in the fluxes that are implied by the atmospheric observations ($z$) but do not exist in any predictor

dataset (i.e., do not match any patterns in $\mathbf{X}\boldsymbol{\beta}$). This term is often referred to as the stochastic component of the fluxes and is

also estimated as part of the GIM. This stochastic component ($\boldsymbol{\zeta}$) can have a variety of spatial or temporal patterns. However,

its structure is represented by a covariance matrix, termed $\mathbf{Q}$ ($m \times m$), where $\boldsymbol{\zeta} \sim \mathcal{N}(\mathbf{0}, \mathbf{Q})$ (e.g., Kitanidis and Vomvoris,

1983; Michalak et al., 2004).

Note that it has become standard practice in GIM studies to estimate the fluxes ($s$) at the highest spatial and temporal

resolution possible (e.g., Gourdji et al., 2010). This setup accounts for small-scale variability in surface fluxes down to the

resolution of the atmospheric model, thereby yielding a more accurate flux estimate and more realistic uncertainty bounds

(e.g., Gourdji et al., 2010). However, the number of unknown fluxes ($s$) usually far exceeds the number of observations ($z$),

and the inverse problem of estimating the fluxes from the data is typically underdetermined, meaning that multiple solutions

consistent with the data are possible (e.g., Mueller et al., 2008; Gourdji et al., 2012; Miller et al., 2013, 2016; Shiga et al.,





2018). To address this issue, the geostatistical approach uses additional information to help determine the structure of the fluxes ($s$). In particular, rather than choosing a diagonal prior covariance matrix $\mathbf{Q}$, which ignores spatiotemporal interactions, it is standard practice to include nonzero off-diagonal elements. These elements guide the spatial and temporal structure of the flux estimate and help interpolate fluxes in locations without a perfect data constraint. Kitanidis (1997) and Wackernagel (2003) review different approaches to modeling these off-diagonal elements.

The geostatistical approach then uses Bayes formula to derive the posterior distribution $p(\mathbf{s}, \boldsymbol{\beta}|\mathbf{z})$ as

$$p(\mathbf{s}, \boldsymbol{\beta}|\mathbf{z}) \propto p(\mathbf{z}|\mathbf{s}, \boldsymbol{\beta})p(\mathbf{s}|\boldsymbol{\beta})p(\boldsymbol{\beta}) \tag{4}$$

where $\propto$ denotes proportionality. The expressions for the likelihood $p(\mathbf{z}|\mathbf{s}, \boldsymbol{\beta})$ can be derived from Eq. 1 and Eq. 2, and the prior $p(\mathbf{s}|\boldsymbol{\beta})$ can be derived from Eq. 3. Furthermore, taking $p(\boldsymbol{\beta}) \propto 1$, we can derive

$$p(\mathbf{s}, \boldsymbol{\beta}|\mathbf{z}) \propto \exp\left(-\tfrac{1}{2}(\boldsymbol{z} - \mathbf{H}\boldsymbol{s})^T\mathbf{R}^{-1}(\boldsymbol{z} - \mathbf{H}\boldsymbol{s}) - \tfrac{1}{2}(\boldsymbol{s} - \mathbf{X}\boldsymbol{\beta})^T\mathbf{Q}^{-1}(\boldsymbol{s} - \mathbf{X}\boldsymbol{\beta})\right) \tag{5}$$

The best estimate of the fluxes can be computed by maximizing $p(\mathbf{s}, \boldsymbol{\beta}|\mathbf{z})$, the posterior distribution; alternatively, it can be

obtained by minimizing the negative logarithm of the posterior density which yields the equation (e.g., Kitanidis and Vomvoris, 1983; Michalak et al., 2004):

$$L(\boldsymbol{s}, \boldsymbol{\beta}) = \tfrac{1}{2}(\boldsymbol{z} - \mathbf{H}\boldsymbol{s})^T\mathbf{R}^{-1}(\boldsymbol{z} - \mathbf{H}\boldsymbol{s}) + \tfrac{1}{2}(\boldsymbol{s} - \mathbf{X}\boldsymbol{\beta})^T\mathbf{Q}^{-1}(\boldsymbol{s} - \mathbf{X}\boldsymbol{\beta}) \tag{6}$$

Both the fluxes ($s$) and the coefficients ($\boldsymbol{\beta}$) are unknown in this equation and are estimated as part of the GIM. Typically, the number of coefficients ($\boldsymbol{\beta}$) is relatively small (e.g., $< 1 \times 10^2$), but the number of unknown fluxes can be very large (e.g.,

$> 1 \times 10^6$) (e.g., Gourdji et al., 2012; Miller et al., 2016; Shiga et al., 2018). The next section reviews a direct approach for minimizing of the function in Eq. 6 and solving the GIM.

## 3   Direct approach to solving the GIM and associated challenges for large datasets

The classical solution to the GIM requires solving a single system of linear equations (e.g., Kitanidis, 1996; Saibaba and Kitanidis, 2012); the best estimate $\hat{s}$ is obtained as

$$\hat{s} = \mathbf{X}\boldsymbol{\beta} + \mathbf{Q}\mathbf{H}^T\boldsymbol{\xi} \tag{7}$$

where $\boldsymbol{\xi}$ ($n \times 1$) is an unknown vector of weights, and $\boldsymbol{\beta}$ ($p \times 1$) are the unknown coefficients, as in Eq. 6. These vectors are obtained by solving a linear system of equations:

$$\begin{bmatrix} \mathbf{HQH}^T + \mathbf{R} & \mathbf{HX} \\ (\mathbf{HX})^T & \mathbf{0} \end{bmatrix} \begin{bmatrix} \boldsymbol{\xi} \\ \boldsymbol{\beta} \end{bmatrix} = \begin{bmatrix} \boldsymbol{z} \\ \mathbf{0} \end{bmatrix}. \tag{8}$$

Note that there are several equivalent sets of equations for estimating the fluxes (e.g., Michalak et al., 2004), and the set of

equations shown above is commonly referred to as the *dual function form*. When this linear system is solved using a direct method, such as Gaussian elimination, or LU factorization, we refer to it as the direct solution.



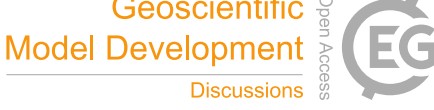

Over the past decade, the in situ and satellite greenhouse gas observation networks have expanded and so have the dimensions of many inverse problems, resulting in major computational costs associated with solving Eq. 8. The first issue involves computing the prior covariance matrix $\mathbf{Q}$, which is affected by the number of unknown fluxes ($m$). This number is typically the product of the number of model grid boxes and the number of time periods in the inverse model, and the total number of unknown fluxes can therefore exceed $1 \times 10^6$ even if the number of model grid boxes or the number of time periods is modest

(e.g., Gourdji et al., 2012; Miller et al., 2016; Shiga et al., 2018). The second issue involves multiplying $\mathbf{H}$ and $\mathbf{Q}$, and the third involves solving the linear system in Eq. 8. We examine each of these issues and possible strategies to address them.

The first issue is that the covariance matrix $\mathbf{Q}$ can be too large to store in memory, to invert, and/or to feasibly use in matrix–matrix multiplication if the number of unknown fluxes is large. The covariance matrix $\mathbf{Q}$ is often defined by a relatively small number of parameters (e.g., a variance, a decorrelation length, and a decorrelation time; Gourdji et al., 2012; Miller et al.,

2016), but the matrix is nonetheless large and is often non-sparse (e.g., Yadav and Michalak, 2013).

Several studies develop strategies for solving the GIM system of equations when the number unknown fluxes is large (e.g., Fritz et al., 2009; Yadav and Michalak, 2013; Saibaba and Kitanidis, 2012; Ambikasaran et al., 2013a, b). Many GIM studies circumvent this computational bottleneck by decomposing the covariance matrix $\mathbf{Q}$ into a spatial component and a temporal component (Yadav and Michalak, 2013):

$$\mathbf{Q} = \sigma_Q^2 (\mathbf{D} \otimes \mathbf{E}) \tag{9}$$

$$\mathbf{Q}^{-1} = \frac{1}{\sigma_Q^2} (\mathbf{D}^{-1} \otimes \mathbf{E}^{-1}) \tag{10}$$

where $\sigma_Q^2$ is a parameter that controls the variance in $\mathbf{Q}$, $\mathbf{D}$ describes the temporal covariance, $\mathbf{E}$ describes the spatial covariance, and $\otimes$ is the Kronecker product. One can multiply $\mathbf{Q}$ or $\mathbf{Q}^{-1}$ by a vector or matrix using $\sigma_Q^2$, $\mathbf{D}$, and $\mathbf{E}$ without ever explicitly formulating $\mathbf{Q}$. This approach can dramatically improve computational tractability and is described in detail in Yadav

and Michalak (2013). In this paper, we also model the entries of $\mathbf{Q}$ using a spherical covariance model (e.g., Kitanidis, 1997; Wackernagel, 2003). Unlike other potential choices of covariance models, a spherical model decays to zero at the correlation length and correlation time. This property means that $\mathbf{D}$ and $\mathbf{E}$ will be sparse matrices, saving both memory and computing time. Several recent papers discuss additional strategies if $\mathbf{D}$ and $\mathbf{E}$ still pose a computational bottleneck. These include formulating the components of $\mathbf{Q}$ as a hierarchical matrix or a structured matrix (such as Toeplitz or block Toeplitz) (e.g., Fritz

et al., 2009; Saibaba and Kitanidis, 2012; Ambikasaran et al., 2013a, b).

A second computational bottleneck is the cost of forming the matrix $\mathbf{\Psi} = \mathbf{HQH}^T + \mathbf{R}$. Many atmospheric models do not explicitly formulate $\mathbf{H}$. Instead, these atmospheric models pass a vector through the forward or adjoint model. In other words, they calculate the product of $\mathbf{H}$ or $\mathbf{H}^T$ and a vector. In these cases, the direct solution to the GIM would require thousands or millions of atmospheric model simulations to calculate $\mathbf{HQH}^T$, an approach that would be computationally burdensome and

impractical. Many studies that employ classical Bayesian inverse modeling circumvent this problem by iterating toward the solution in a way that does not require calculating the product $\mathbf{HQ}$ (e.g., Baker et al., 2006; Henze et al., 2007; Meirink et al., 2008). This approach is often referred to as a variational or adjoint-based inverse model (e.g., Brasseur and Jacob, 2017).





A third issue involves solving the linear system in Eq. 8 when the number of measurements ($n$) is large. The direct solution to the GIM requires inverting a matrix of dimensions $(n+p) \times (n+p)$ that is often non-sparse. Greenhouse gas observing satellites can collect millions of observations per year, yielding a matrix that is too large to feasibly invert. The number of available observations will continue to grow as more greenhouse gas observing satellites launch into orbit and as the existing observational record becomes longer.

To solve this third issue, one could reduce the dimensions of the inverse model to the point where it is computationally manageable, but these strategies have drawbacks. First, one could average down the data until $n$ is much smaller, and the matrix inverse required by the direct solution is computationally feasible. However, this strategy can require a large degree of averaging, and one might average over meaningful variability in the observations and/or in the meteorology, reducing the accuracy of the flux estimate. A second option is to break up the inverse model into smaller time periods until the number of data points ($n$) in each time period is manageable. This approach, however, brings several pitfalls. The number of satellite observations can be large even for relatively short time periods. In addition, the stochastic component of the fluxes may covary across long time periods or from one year to another, and it can be important to account these covariances in the inverse model via the covariance matrix $\mathbf{Q}$. Furthermore, one may want to make inferences about the fluxes across longer time periods. For example, inverse modeling studies commonly report estimated total annual carbon budgets and associated uncertainties. A third option is to use ensemble-based approaches (e.g., an ensemble Kalman smoother) (e.g., Chatterjee et al., 2012; Chatterjee and Michalak, 2013). These approaches generally require a large number of ensemble members to accurately reproduce the flux field, and one must run the forward model once for each ensemble member. As a result, this approach can require hundreds or thousands of forward model simulations to produce an estimate that is close to the direct solution, depending upon the dimensions and complexity of the inverse problem (e.g., Chatterjee et al., 2012). In this study, we explore several approaches to GIMs (Eq. 6) that are practical for very large datasets and do not necessarily require the dimension reduction strategies described above. These approaches are also compatible with atmospheric models that do not explicitly formulate the matrix $\mathbf{H}$ (i.e., the variational or adjoint-based approach).

## 4 Approaches to calculating the best estimate

The direct solution is often intractable for inverse problems with large datasets. An alternate is to use an iterative or variational approach to reach the best estimate of the fluxes, and we discuss two approaches. Each uses a different form of the GIM equations and uses a different type of numerical solver to iterate toward the best estimate.

### 4.1 Quasi-Newton approach

Many existing variational or adjoint-based inverse modeling studies use an iterative, quasi-Newton approach to estimate the fluxes (e.g., Baker et al., 2006; Henze et al., 2007). This strategy requires creating scripts that calculate the cost function (Eq. 6) and gradient, and the quasi-Newton solver will iterate toward the solution using these two inputs. The most common quasi-Newton solver in existing studies is the Limited-memory Broyden-Fletcher-Goldfarb-Shanno algorithm (L-BFGS) (Nocedal,





1980; Liu and Nocedal, 1989). Furthermore, one variant of this algorithm, L-BFGS-B, will estimate the fluxes ($s$) subject to a bound. For example, some trace gases do not have large surface sinks, and L-BFGS-B can ensure that the estimated fluxes are non-negative. Refer to Miller et al. (2014) for a full discussion of strategies to enforce bounds on atmospheric inverse problems.

The implementation of L-BFGS or L-BFGS-B for a GIM is more complicated than in a classical Bayesian inversion. Specifically, the goal is to use one of these algorithms to estimate the fluxes ($s$), but the cost function in the GIM has an additional unknown variable ($\beta$, Eq. 6). We substitute the equation for the unknown coefficients ($\beta$) into the GIM cost function (Eq. 6),
5 thereby removing $\beta$ (e.g., Kitanidis, 1995):

$$L(s) = \tfrac{1}{2}(z - \mathbf{H}s)^T \mathbf{R}^{-1}(z - \mathbf{H}s) + \tfrac{1}{2}s^T \mathbf{G}s \tag{11}$$

$$\mathbf{G} = \mathbf{Q}^{-1} - \mathbf{Q}^{-1}\mathbf{X}(\mathbf{X}^T\mathbf{Q}^{-1}\mathbf{X})^{-1}(\mathbf{Q}^{-1}\mathbf{X})^T \tag{12}$$

where $\mathbf{G}$ has dimensions $m \times m$. Note that we never formulate $\mathbf{G}$ explicitly when calculating Eq. 11 because it is often a large, non-sparse $m \times m$ matrix. Rather, we successively multiply $s$ by the individual components of Eq. 12 to avoid formulating a
10 full $m \times m$ matrix. In addition, we use the Kronecker product identity in Eq. 10 to avoid explicitly formulating $\mathbf{Q}^{-1}$. The SI describes this approach in greater detail.

We can further speed up the convergence of L-BFGS using a variable transformation. We transform the fluxes from $s$ to $s^*$, similar to the approach used in a handful of atmospheric inverse modeling studies (e.g., Baker et al., 2006; Meirink et al., 2008). The strategy is to first solve a transformed optimization problem for $s^*$, and then to obtain $s$ using the relations:

$$s^* = \mathbf{Q}^{-\frac{1}{2}}s \tag{13}$$

$$s = \mathbf{Q}^{\frac{1}{2}}s^* \tag{14}$$

$$\mathbf{Q}^{\frac{1}{2}} = \sigma_Q(\mathbf{D}^{\frac{1}{2}} \otimes \mathbf{E}^{\frac{1}{2}}) \tag{15}$$

where $\mathbf{Q}^{\frac{1}{2}}$ is the symmetric square root of $\mathbf{Q}$. Note that we never explicitly formulate $\mathbf{Q}^{\frac{1}{2}}$ but instead do all matrix operations on the individual components of $\mathbf{Q}^{\frac{1}{2}}$. We then substitute the equation for $s^*$ into the cost function ($L_s$):

$$L(s^*) = \tfrac{1}{2}(z - \mathbf{H}\mathbf{Q}^{\frac{1}{2}}s^*)^T \mathbf{R}^{-1}(z - \mathbf{H}\mathbf{Q}^{\frac{1}{2}}s^*) + \tfrac{1}{2}s^{*T}\mathbf{G}^*s^* \tag{16}$$

$$\nabla L(s^*) = -\tfrac{1}{2}\mathbf{Q}^{\frac{1}{2}}\mathbf{H}^T\mathbf{R}^{-1}(z - \mathbf{H}\mathbf{Q}^{\frac{1}{2}}s^*) + \tfrac{1}{2}\mathbf{G}^*s^* \tag{17}$$

$$\mathbf{G}^* = \mathbf{I} - \mathbf{Q}^{-\frac{1}{2}}\mathbf{X}(\mathbf{X}^T\mathbf{Q}^{-1}\mathbf{X})^{-1}\mathbf{X}^T\mathbf{Q}^{-\frac{1}{2}} \tag{18}$$

The functions $L_{s^*}$ and $\nabla L(s^*)$ are then used as inputs to the L-BFGS algorithm, and the resulting optimization can converge in fewer iterations than using the cost function without the transformation (Eq. 11). Sect. 7 includes further discussion of why
25 L-BFGS without a transformation may converge slowly and how a variable transformation can remedy the problem.

### 4.2 Minimum residual approach

The minimum residual approach described here uses a very different strategy for the iterative optimization (Paige and Saunders, 1975). The quasi-Newton approach described above will search for the minimum of the cost function with the help of the





gradient (Eq. 16-17). By contrast, there is a class of solvers that will estimate the solution to a linear system of equations where
one side is too large to invert, and this class of solvers offers an alternative strategy. Specifically, these methods will solve a
system of equations of the form $\mathbf{A}\boldsymbol{x} = \boldsymbol{b}$ where $\boldsymbol{x}$ is a vector of unknown values, $\boldsymbol{b}$ is a known vector, and $\mathbf{A}$ is a matrix that is
too large to store in memory or too expensive to form explicitly. This strategy has been employed by inverse modeling studies
in hydrology and seismology (Saibaba and Kitanidis, 2012; Saibaba, A.K. et al., 2012; Liu et al., 2014b; Lee et al., 2016).

This strategy can also be employed to estimate the fluxes using Eq. 8. We cannot solve these equations directly for inverse
problems with large datasets (i.e., large $n$); the left hand side of Eq. 8 becomes too large to invert and too expensive to explicitly
form. Instead, we use the minimum residual method to estimate $\boldsymbol{\xi}$ and $\boldsymbol{\beta}$ (e.g., Barrett et al., 1994). This class of algorithms
only require a function that will calculate the left hand side of Eq. 8 given some guess for $\boldsymbol{\xi}$ and $\boldsymbol{\beta}$. Then, one can iteratively
compute an approximate solution by forming a series of matrix-vector products – the product of $\boldsymbol{\xi}$ and $\boldsymbol{\beta}$ with the matrices on
the left hand size of of Eq. 8. This approach also makes it feasible to estimate the fluxes ($\boldsymbol{s}$) even if the atmospheric model
does not explicitly calculate $\mathbf{H}$ or $\mathbf{H}^T$. One can pass the vector $\boldsymbol{\xi}$ through the model adjoint to calculate $\mathbf{H}^T\boldsymbol{\xi}$ and then pass
the vector $\mathbf{Q}\mathbf{H}^T\boldsymbol{\xi}$ through the forward model to calculate $\mathbf{H}(\mathbf{Q}\mathbf{H}^T\boldsymbol{\xi})$. As a result, one never needs to calculate or store the
$(n+p) \times (n+p)$ matrix on the left hand side of Eq. 8 or invert that matrix.

Note that some studies use a preconditioner to help the minimum residual algorithm converge more quickly to a solution
(e.g., Saibaba and Kitanidis, 2012; Liu et al., 2014b; Lee et al., 2016). The preconditioner can speed up convergence by reducing
the condition number and/or clustering the eigenvalues near 1. Saibaba and Kitanidis (2012) detail one possible strategy for
preconditioning the GIM, and this approach has subsequently been employed in several studies (e.g., Liu et al., 2014b; Lee
et al., 2016). We do not detail the implementation here, but Saibaba and Kitanidis (2012) describe the step-by-step procedure.
The preconditioner detailed in that study can dramatically speed up convergence (Sect. 7) but may require tens to thousands of
forward model runs as a precomputational cost to realize these improvements in convergence, depending upon the complexity
of the covariance matrix $\mathbf{Q}$. Note that these forward model runs can be done simultaneously in parallel. A discussion on the
computational cost is provided in Sect. 7.

## 5  Uncertainty estimation

It is often not possible to estimate the full, posterior covariance matrix when the number of observations and the number of un-
known fluxes are large. Two very different general approaches are often used in atmospheric and hydrologic data assimilation,
and we discuss both here in the context of GIMs. One entails creating an ensemble of randomized simulations or realizations.
The other uses a low-rank approximation of one matrix in the posterior covariance calculations, and this approximation makes
the overall calculations computationally feasible.

### 5.1  Conditional realizations or simulations

One approach to estimate the posterior uncertainties is to generate conditional realizations or Monte Carlo simulations, and
several variational or adjoint-based studies of greenhouse gases have employed this strategy (e.g., Chevallier et al., 2007; Liu





et al., 2014a; Bousserez et al., 2015). A conditional realization is a random sample from the posterior distribution (e.g., Kitanidis, 1995; Michalak et al., 2004; Chevallier et al., 2007), and the statistics of many conditional realizations will approximate the posterior variances and covariances. These variances and covariances can have complex spatial and temporal patterns, and it can be challenging to adequately sample the tails of the posterior distribution. As a result, several studies of atmospheric trace gases use thousands of realizations to sample the posterior distribution and approximate the uncertainties (e.g., Rigby

et al., 2011; Ganesan et al., 2014). In other cases, computational constraints make it impractical to generate more than tens or hundreds of realizations, especially for very large variational or adjoint-based inverse problems (e.g., Chevallier et al., 2007; Liu et al., 2014a).

One approach for generating a single conditional realization is similar to the procedure for calculating the best estimate of the fluxes. To generate a conditional realization, we first generate random samples from $\mathcal{N}(\mathbf{0}, \mathbf{R})$ and $\mathcal{N}(\mathbf{0}, \mathbf{Q})$ and use these

samples to perturb the data and the parameters respectively; we then solve the GIM using any of the three approaches described previously. In each case, the equations are slightly different, and we therefore do not list all of the equations here. Michalak et al. (2004) and Saibaba and Kitanidis (2012) describe how to generate conditional realizations using the direct approach in Sect. 3, Kitanidis (1995) and Snodgrass and Kitanidis (1997) and the SI describe how to generate realizations using quasi-Newton methods (Sect. 4.1), and Kitanidis (1996) and Saibaba and Kitanidis (2012) describe how to generate realizations

using the minimum residual approach (Sect. 4.2).

### 5.2 Reduced rank approach

A number of studies reduce the computational burden of uncertainty estimation by replacing one matrix in the calculations with a low rank approximation. This strategy is based upon the direct calculation of the posterior covariance matrix ($\mathbf{V}_{\hat{\mathbf{s}}}$) (e.g., Saibaba and Kitanidis, 2015):

$$\mathbf{V}_{\hat{\mathbf{s}}} = \mathbf{V}_1 + \mathbf{V}_2\mathbf{V}_3\mathbf{V}_2^T \tag{19}$$

$$\mathbf{V}_1 = (\mathbf{Q}^{-1} + \mathbf{H}^T\mathbf{R}^{-1}\mathbf{H})^{-1} \tag{20}$$

$$\mathbf{V}_2 = \mathbf{V}_1\mathbf{Q}^{-1}\mathbf{X} \tag{21}$$

$$\mathbf{V}_3 = (\mathbf{X}^T\mathbf{Q}^{-1}\mathbf{X} - (\mathbf{Q}^{-1}\mathbf{X})^T\mathbf{V}_1\mathbf{Q}^{-1}\mathbf{X})^{-1} \tag{22}$$

where $\mathbf{V}_{\hat{\mathbf{s}}}$ and $\mathbf{V}_1$ have dimensions $m \times m$, $\mathbf{V}_2$ dimensions $m \times p$, and $\mathbf{V}_3$ dimensions $p \times p$. Note that $\mathbf{V}_1$ is the posterior

covariance matrix in a classical Bayesian inversion. The uncertainty calculations in a GIM include an additional term, notably $\mathbf{V}_2\mathbf{V}_3\mathbf{V}_2^T$. This term accounts for the effect of uncertain coefficients ($\boldsymbol{\beta}$) on the estimated fluxes. These coefficients are estimated as part of the GIM, and uncertainty in these coefficients contributes additional uncertainty to the posterior flux estimate ($\hat{\mathbf{s}}$). Also note that there are many equivalent equations for calculating the posterior covariance matrix, and the form in Eq. 19 is particularly conducive to a low-rank approximation strategy.

This direct calculation of the posterior covariance matrix is not tractable for large inverse problems, in part, because it requires inverting the matrix sum in $\mathbf{V}_1$. Furthermore, it requires computing the matrix-matrix product $\mathbf{H}^T\mathbf{R}^{-1}\mathbf{H}$, a step that can be impractical for models that do not explicitly formulate $\mathbf{H}$.



Instead, we use a specific low-rank approximation in the calculations for $\mathbf{V}_1$. In other words, we approximate a matrix using a limited number of eigenvectors and eigenvalues, thereby making the calculations for $\mathbf{V}_1$ computationally feasible. A number of studies discuss this strategy in the context of a classical Bayesian inversion (e.g., Meirink et al., 2008; Flath et al., 2011; Spantini et al., 2015). Here, we review how this approach can be applied in the context of a GIM (e.g., Saibaba and Kitanidis,

2015). We summarize the procedure here, but refer to Flath et al. (2011), Spantini et al. (2015), and Saibaba and Kitanidis (2015) additional detail. The main idea is to consider the matrix

$$\mathbf{Q}^{\frac{1}{2}}\mathbf{H}^{T}\mathbf{R}^{-1}\mathbf{H}\mathbf{Q}^{\frac{1}{2}} \tag{23}$$

where $\mathbf{Q}^{\frac{1}{2}}$ can be the symmetric square root or Cholesky decomposition of $\mathbf{Q}$. This matrix is sometimes referred to as the *prior-preconditioned data-misfit part of the Hessian*. Previous studies have leveraged the fact that, in many applications, this

matrix has rapidly decaying eigenvalues and therefore, can be accurately approximated using a low-rank matrix (e.g., Meirink et al., 2008; Flath et al., 2011; Spantini et al., 2015). This low-rank representation has a dual purpose: it is more efficient to store this matrix in memory and improves the efficiency of computations. There are several methods to compute this low-rank approximation, and these methods can estimate the eigenvalues and vectors without storing the above matrix (Eq. 23) in memory. Instead, these algorithms (e.g., Krylov subspace methods) iteratively estimate the largest eigenvalues and eigenvectors

of a matrix $\mathbf{A}$ given some function that calculates $\mathbf{A}\boldsymbol{x}$, where $\boldsymbol{x}$ is a vector that is provided by the algorithm and does not need to be specified by the user. For example, the `eigs` function in MATLAB offers an interface to these algorithms.

In this study, we estimate the eigenvectors and eigenvalues using a randomized algorithm developed in Halko et al. (2011). This approach requires just over two forward model runs and two adjoint model runs per (approximate) eigenpair, and this algorithm is therefore less computationally intensive than many other available algorithms. Furthermore, these forward and

adjoint model runs can be generated in parallel, reducing the required computing time. Using this approach, we obtain the low-rank approximation

$$\mathbf{Q}^{\frac{1}{2}}\mathbf{H}^{T}\mathbf{R}^{-1}\mathbf{H}\mathbf{Q}^{\frac{1}{2}} \approx \mathbf{U}\boldsymbol{\Lambda}\mathbf{U}^{T} \tag{24}$$

where $\mathbf{U}$ $(n \times \ell)$ contains the approximate eigenvectors, $\boldsymbol{\Lambda}$ $(\ell \times \ell)$ is a diagonal matrix whose diagonals contain the approximate eigenvalues and $\ell$ is the number of approximate eigenpairs computed. Note that Saibaba and Kitanidis (2015) and Saibaba et al.

(2016) provide an alternative approach that does not require taking the symmetric square root or Cholesky decomposition of $\mathbf{Q}$. That approach is a good choice if the symmetric square root or Cholesky decomposition of $\mathbf{D}$ and/or $\mathbf{E}$ is difficult to compute.

We can use these eigenvalues/vectors to approximate $\mathbf{V}_1$ using the Woodbury matrix identity (Flath et al., 2011; Spantini et al., 2015):

$$\mathbf{V}_1 \approx (\mathbf{Q} - \mathbf{Q}^{\frac{1}{2}}\mathbf{U}\boldsymbol{\Lambda}(\mathbf{I} + \boldsymbol{\Lambda})^{-1}\mathbf{U}^{T}\mathbf{Q}^{\frac{1}{2}}) \tag{25}$$

The more eigenvectors and eigenvalues used in Eq. 25, the better the approximation. This approximation can subsequently be plugged into the expression for $\mathbf{V}_{\hat{\boldsymbol{s}}}$ to obtain an approximation to the posterior covariance matrix.

This reduced rank approach greatly improves the tractability of posterior uncertainty calculations, but one computational road block remains; $\mathbf{V}_{\hat{\boldsymbol{s}}}$ is usually too large to store in memory. However, one can calculate uncertainties for individual grid





boxes or for aggregate regions without storing $\mathbf{V_{\hat{s}}}$. To do so, multiply the right-hand side of Eq. 19 by a vector, resulting in a series of matrix–vector calculations. In this case, the vector should have a one in the flux grid box(es) of interest and a zero elsewhere. If that vector has a one for multiple elements, then the calculation will yield the uncertainty in the flux estimate summed across several model grid boxes. For example, the following equation will compute the uncertainty in the total flux summed over all locations and times:

$$\mathbf{1}^T\mathbf{V}_1\mathbf{1} + \mathbf{1}^T(\mathbf{V}_2(\mathbf{V}_3(\mathbf{V}_2^T\mathbf{1}))) \tag{26}$$

where $\mathbf{1}$ ($m \times 1$) is a vector of ones. Note that it may be necessary to convert the units of the fluxes in the course of these calculations. For example, atmospheric models that use a latitude–longitude grid will have grid boxes with different areas, and it may be necessary to multiply the vector $\mathbf{1}$ by the area of each grid box to account for these differing grid box areas.

## 6 Case study from the OCO-2 satellite

We evaluate the inverse modeling algorithms described in this paper using two case studies based on NASA's OCO-2 satellite. OCO-2 was launched in September 2014 and is NASA's first satellite dedicated to observing atmospheric $CO_2$ from space. This section provides an overview of the case studies while the Supplement provides additional, detailed descriptions.

The first case study is small enough such that it can be solved using the direct approach. We can therefore use it to compare the iterative GIM algorithms against the direct solution. The second case study, by contrast, is too large for a direct solution but is indicative of the typical size of the datasets encountered in satellite-based inverse modeling. In the first case study, we estimate six weeks of $CO_2$ fluxes across terrestrial North America from late June through July 2015; we use a total of $1.92 \times 10^4$ synthetic OCO-2 observations to estimate $1.05 \times 10^6$ unknown $CO_2$ fluxes. In the latter case study, we estimate a full year of $CO_2$ fluxes (Sept. 2014 - Aug. 2015) for the same geographic domain, a total of $9.88 \times 10^4$ synthetic observations and $9.41 \times 10^6$ unknown $CO_2$ fluxes.

We build these case studies using a synthetic data setup; we model $XCO_2$ using $CO_2$ fluxes from CarbonTracker (CT2017) (Peters et al., 2007; NOAA Global Monitoring Division, 2019) and an atmospheric transport model, add noise to the model outputs to simulate measurement and model errors, and finally estimate $CO_2$ fluxes using these synthetically generated $XCO_2$ observations. We specifically estimate the fluxes at a 3-hourly temporal resolution and a $1° \times 1°$ latitude-longitude resolution. The atmospheric transport simulations used here are from NOAA's CarbonTracker-Lagrange program (e.g., Hu et al., 2019; NOAA Global Monitoring Division, 2019) and were generated using the Weather Research and Forecasting (WRF) Stochastic Time-Inverted Lagrangian Transport Model (STILT) modeling system (e.g., Lin et al., 2003; Nehrkorn et al., 2010) .

Note that one could also pair the inverse modeling algorithms here with an adjoint-based model that does not produce an explicit $\mathbf{H}$ matrix. However, WRF-STILT produces an explicit $\mathbf{H}$ matrix, making it straightforward to evaluate each GIM algorithm against the direct solution.

We also use a non-informative deterministic model ($\mathbf{X}\boldsymbol{\beta}$) in the case studies. In this setup, the matrix $\mathbf{X}$ only consists of columns of ones. As a result, any spatial or temporal patterns in the fluxes are only the result of the observational constraint and are not the reflection of any prior flux estimate.





In addition, the covariance matrix $\mathbf{Q}$ includes both spatial and temporal covariances. We estimate the spatial and temporal
properties (e.g., variances and covariances) of CT2017 fluxes for 2014-2015 and use those properties to populate $\mathbf{Q}$. The
covariance matrix $\mathbf{R}$ is diagonal for the setup here, and we set the diagonal values at $(2\ \text{ppm})^2$. This value is comparable to the
combined model and data errors estimated in (Miller et al., 2018). That study included a detailed error analysis using OCO-2
observations and atmospheric model simulations from the same time period as this study.

## 7   Discussion

### 7.1   Best estimate of the fluxes

We test out several algorithms for estimating the fluxes ($s$), and all but one of these algorithms converges quickly toward the
solution. The minimum residual approach (with and without a preconditioner) and the L-BFGS with a variable transformation
converge quickly for both case studies (Figs. 1, 2, and 3). Note that we cannot compare the larger case study to a direct solution,
but we do compare the results against $CO_2$ fluxes estimated using a very large number of iterations (250 in this case).

By contrast, the L-BFGS algorithm without a variable transformation converges very slowly to the solution (Figs 1 and 3).
The L-BFGS algorithm makes a simple approximation for the posterior covariance matrix at each iteration (i.e., the inverse
Hessian); it uses this approximation and the gradient of the cost function to determine the direction of steepest descent and
iterate toward the solution (e.g., Nocedal, 1980; Liu and Nocedal, 1989). Most L-BFGS algorithms use a memory-saving sparse
matrix like the identity matrix as an initial guess for the inverse Hessian. In the case studies here, the inverse Hessian without
the variable transformation has complex off-diagonal elements, and the identity matrix is therefore a poor approximation. By
contrast, the inverse Hessian with the variable transformation has relatively small off-diagonal elements, and the identity matrix
is a much better approximation. This difference likely explains why the L-BFGS algorithm without the variable transformation
converges far more slowly than the algorithm with the transformation.

Of all algorithms, the minimum residual algorithm with preconditioning converges in the fewest number of iterations (Fig.
1). However, this faster convergence comes at a cost. We test out a preconditioner with $\ell = 400$ and $\ell = 2500$ approximate
eigenvectors. Only the preconditioner with 2500 eigenvectors yields faster convergence and only in the smaller of the two case
studies. To construct this preconditioner, we had to apply each of the 2500 eigenvectors to the forward model ($\mathbf{H}$). These for-
ward model runs may be achievable if an explicit $\mathbf{H}$ matrix is available, if the forward model is not computationally expensive,
and/or if a large computing cluster is available to distribute these runs across many cores. However, there are many instances
when it may be impractical to generate such a large number of forward model runs. Since the preconditioner can be used for
computing the best estimate as well as generating the conditional realizations, the upfront cost of constructing the precondi-
tioner may be reutilized in the uncertainty quantification step in the following ways: (1) the cost can be amortised if hundreds
or thousands of conditional realizations are generated, and (2) information collected during the construction can also be used
in the reduced-rank computations. Moreover, the use of the randomized algorithm for constructing the preconditioners can be
parallelized across the forward model runs. It may also be possible to develop a more computationally-efficient preconditioner,
but that objective is beyond the scope of this study.





It is also important to note that these algorithms converge more quickly on an accurate flux estimate for aggregate space and time scales. In this study, we always estimate the fluxes at a 3-hourly time resolution. After estimating the fluxes, we average the estimate within each grid box across the entire month or year and compare this time-averaged flux to the direct solution (Figs. 1 and 3). These monthly and annual averages are comparable to the direct solution after only a few iterations of the L-BFGS or minimum residual algorithms. By contrast, it takes more iterations for each algorithm to converge on a flux estimate that is accurate relative to the direct solution for each 3-hour time interval. Many GIM studies of $CO_2$ report monthly flux totals (e.g., Gourdji et al., 2012; Shiga et al., 2018), and these monthly totals may therefore be a more important quantity to robustly estimate than 3-hourly fluxes.

Furthermore, the number of iterations required to converge on the solution does not appear to change dramatically with the size of the problem (Figs 1 and 3). Most algorithms converge for aggregate time periods (Fig. 1b) after 50 iterations in the 6-week case study and after about 75 iterations in the year-long case study (Fig. 3b).

Ultimately, the best or optimal algorithm will likely depend upon the specifics of the inverse modeling problem in question. For example, several existing variational or adjoint-based inverse models are already built to work with the L-BFGS algorithm, and it may be more convenient to adapt this existing infrastructure to the geostatistical approach (e.g., Baker et al., 2006; Henze et al., 2007). By contrast, the minimum residual approach does not require inverting or decomposing the covariance matrix components (e.g., $\mathbf{R}$, $\mathbf{D}$, $\mathbf{E}$). This strategy may be advantageous if $\mathbf{R}$ contains off-diagonal elements and is therefore difficult to invert. Similarly, this strategy may be advantageous if either $\mathbf{D}$ or $\mathbf{E}$ is too large to invert or decompose. The choice of whether or not to use a preconditioner with minimum residual approach also depends upon the computational burden of passing many vectors through the forward model. However, the development of new preconditioning strategies could alter this trade-off.

## 7.2 Uncertainty quantification

The reduced rank approach produces the most conservative uncertainty estimates; it will typically approximate uncertainties that are equal to or larger than the true uncertainties. By contrast, the conditional realizations will typically underestimate the posterior uncertainties. Figures 4 and 5 show the uncertainties estimated for both case studies as a function of the number of forward or adjoint model runs. Both approaches converge toward the true posterior uncertainties as the number of eigenvectors or conditional realizations increases.

These approaches will tend to under- and overestimate the uncertainties due to the approximations involved in each. Conditional realizations randomly sample from the posterior uncertainties, and one may need to generate hundreds or thousands of realizations to effectively sample the entire uncertainty space, particularly for large, complex inverse problems. By contrast, the reduced rank approach approximates the posterior uncertainties by subtracting an update from the prior covariance matrix ($\mathbf{Q}$) (Eq. 25). This update will always be too small if it is approximated using a limited number of eigenvectors and eigenvalues, yielding posterior uncertainties that are too large.

Neither of these approaches provides a silver bullet, so to speak, for estimating the uncertainties. Both require a large number of forward and adjoint model runs, a requirement that can be computationally intensive, depending on the requirements for the forward and adjoint models. These model runs can be generated in parallel for either approach, ameliorating this

computational burden. By contrast, one can produce an uncertainty estimate using a smaller number of model runs, but the

resulting uncertainty estimates will either be too small (using conditional realizations) or too large (using the reduced rank approach). It is arguably more conservative to generate posterior uncertainties that are too large than too small; the latter may cause a modeler to overstate the results or incorrectly conclude that a result is statistically significant when it is not. However, neither of these outcomes is ideal.

## 8   Conclusions

The sheer number of global, atmospheric greenhouse gas observations has grown dramatically with the launch of new satellites and the expansion of in situ monitoring efforts. This article discusses several practical strategies for GIMs when the number of atmospheric observations is large. Specifically, we adapt computational and statistical strategies from a variety of academic disciplines to the problem of estimating greenhouse gas fluxes. We then use $CO_2$ observations from OCO-2 as a lens to evaluate the strengths and weaknesses of each strategy.

We discuss two strategies for generating the best estimate of the fluxes, one that iterates toward the solution using a quasi-Newton approach and the other using a minimum residual approach. Both strategies provide feasible options for estimating millions of unknown fluxes using large satellite or in situ atmospheric datasets. Both can be paired with different types of atmospheric transport models, including particle trajectory models like STILT or gridded Eulerian models that do not save an explicit adjoint matrix. The choice between these two approaches likely depends upon the specifics of the inverse problem in

question and the ease of integrating each into any existing model infrastructure or code.

We further explore two possible strategies for approximating the posterior uncertainties – the generation of conditional realizations and a reduced rank approach. Conditional realizations have numerous and varied applications in inverse problems (e.g., Kitanidis, 1997; Michalak et al., 2004), but we do not recommend them as the only means of estimating the posterior uncertainties unless it is tractable to generate hundreds to thousands of realizations. Otherwise, the estimated uncertainties

will likely be too small and provide a misleading level of confidence in the estimated fluxes. The reduced rank approach, by contrast, will not underestimate the posterior uncertainties and therefore provides a more conservative uncertainty estimate that will not overstate the results.

*Code and data availability.*   The code associated with this article is available on GitHub at dx.doi.org/10.5281/zenodo.3241524. This code repository includes general scripts that can be applied to a variety of inverse modeling problems and scripts that will specifically run the first case study described in this article. Furthermore, the input files for the first case study are available on Zenodo at dx.doi.org/10.5281/zenodo.3241466.

*Author contributions.*   S.M.M. and A.K.S. designed and wrote the study. S.M.M. conducted the analysis. M.E.T., M.E.M., and A.E.A. designed and created the WRF-STILT model simulations used in the study.



*Competing interests.* The authors declare they have no competing interests.

*Acknowledgements.* We thank Zichong Chen (Johns Hopkins University) and Thomas Nehrkorn (Atmospheric And Environmental Research, Inc.) for their help with the article. This work is funded by NASA ROSES grant no. 80NSSC18K0976 and partially supported by NSF grant no. DMS-1720398. CarbonTracker CT2017 results are provided by NOAA ESRL, Boulder, Colorado, USA from the website at http: //carbontracker.noaa.gov.





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



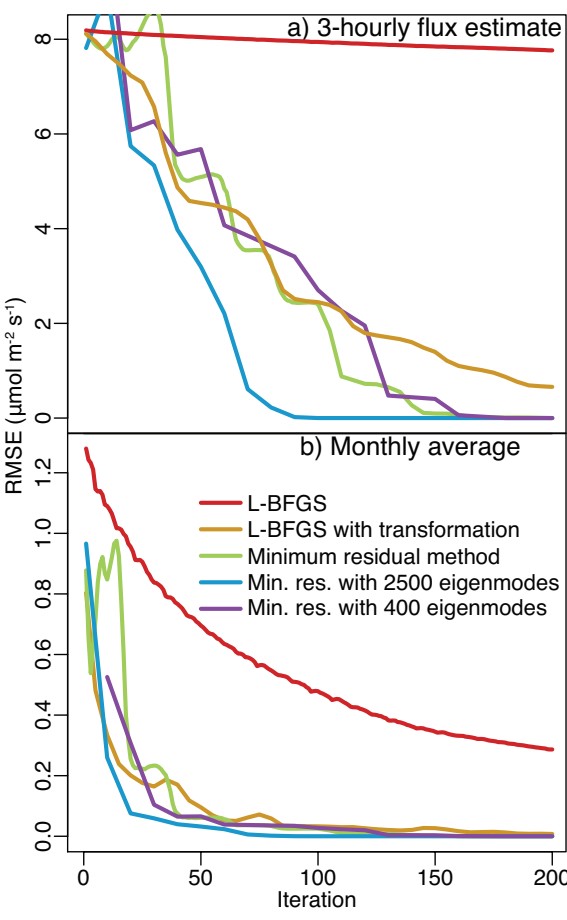

**Figure 1.** Root mean squared error (RMSE) relative to the direct solution for the six week case study. The top panel (a) displays RMSE calculated using each of the $1.05 \times 10^6$ flux grid boxes. The minimum residual method with a preconditioner converges most quickly on the direct solution while L-BFGS shows poor convergence. We also average the 3-hourly posterior flux estimate to obtain a monthly-averaged flux for each model grid box, and the error in this monthly average is displayed in panel (b). The flux estimate, when averaged up to this aggregate monthly scale, converges more quickly to the direct solution than the individual 3-hourly fluxes.





**Figure 2.** The "true" $CO_2$ fluxes from CT2017 used in the 6-week case study (a) and the direct estimate of the fluxes (b). We estimate 3-hourly fluxes but average them across the month of July in this figure. Panels (c-e) display the flux estimate from L-BFGS as a function of iteration number, panels (f-h) from L-BFGS with a variable transformation, and panels (i-k) from the minimum residual approach. Both L-BFGS with the transformation and the minimum residual approach resemble the direct solution after about 20 iterations (panels d, g, and j).





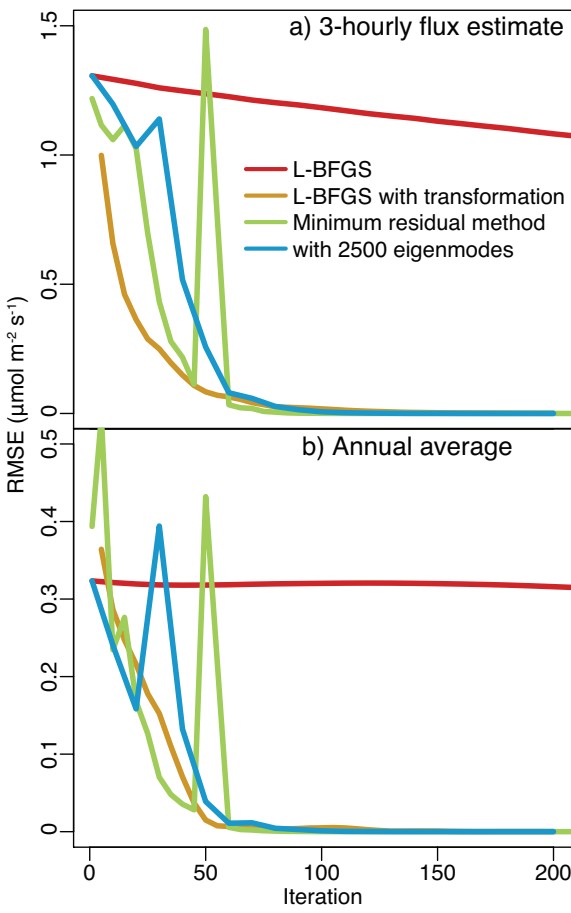

**Figure 3.** RMSE relative to the direct solution for the full year case study. The top panel (a) displays RMSE calculated using each of the $9.4 \times 10^6$ fluxes. We also average the 3-hourly posterior flux estimate to obtain a annually-averaged flux for each model grid box (b). The full year case study is much larger than the 6-week case study (i.e., includes more observations and unknowns), yet the flux estimate converges to the solution after a similar number of iterations.





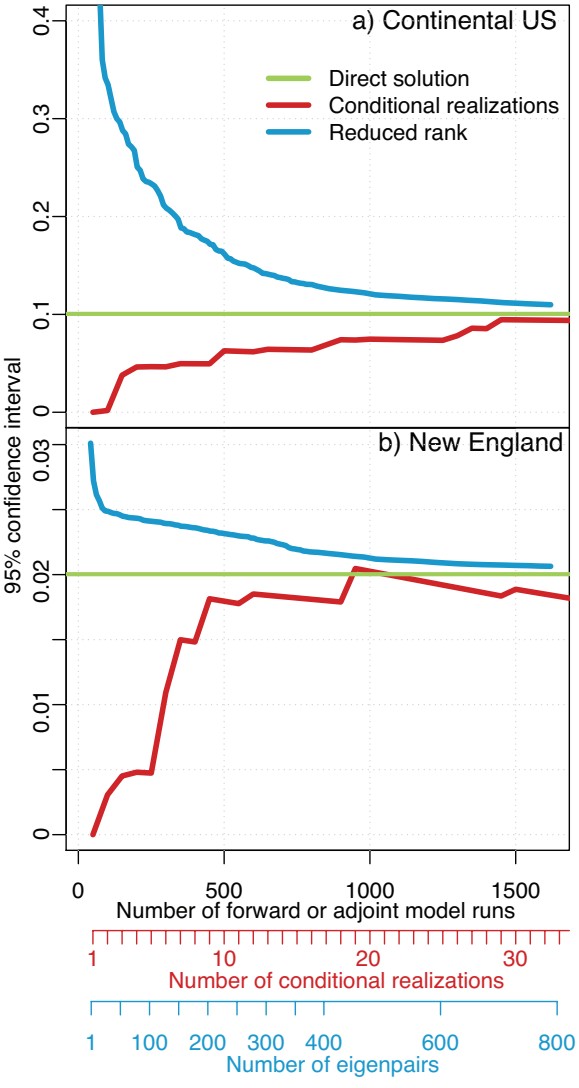

**Figure 4.** Estimated uncertainties using conditional realizations and the reduced rank approach for the six week case study. Panel (a) shows the estimated uncertainty in the total estimated $CO_2$ flux for the US and panel (b) for New England. Both approaches converge on the direct estimate as the number of realizations and eigenpairs increase. Note that we estimate the uncertainties for each model grid box and each 3-hour time period, but we report the uncertainties for broader aggregate regions in this figure. Most existing GIM studies report the uncertainties for large regions because these regions are either more ecologically or policy relevant than the individual model grid boxes. Additionally, the uncertainties are almost always smaller proportional to the total flux for large regions than for individual grid boxes; atmospheric observations usually provide a much more robust constraint on regional flux totals than for individual model grid boxes.





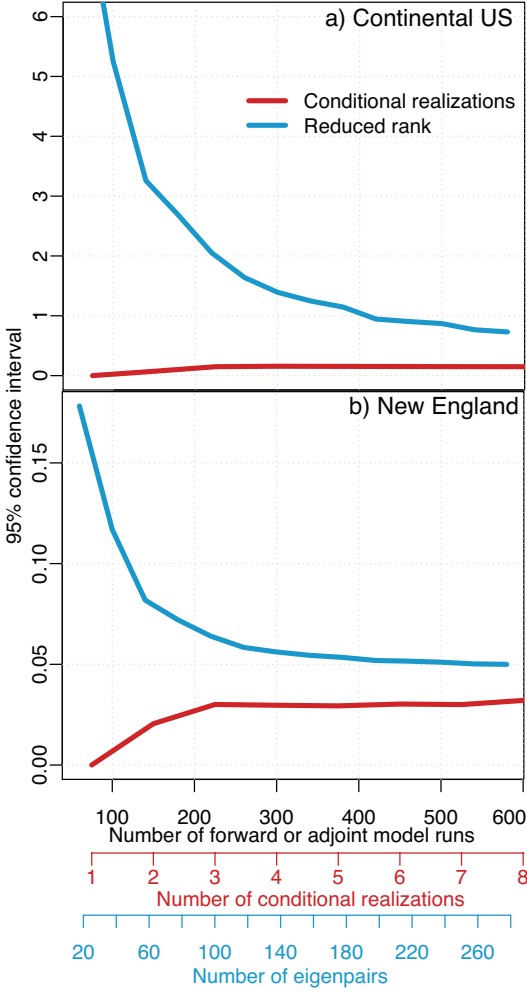

**Figure 5.** Estimated uncertainties for the full year case study, analogous to Fig. 4. The uncertainties estimated by the reduced rank approach are consistently higher than the conditional realizations, though they begin to converge as the number of realizations and eigenpairs increases. Note that it is not possible to obtain a direct estimate for the uncertainties given the size of this case study, so we cannot evaluate the accuracy of the uncertainty estimates shown here.