# Peer review of "Geostatistical inverse modeling with very large datasets: an example from the OCO-2 satellite"

_Geoscientific Model Development, 2019_

## Referee Comment (RC1) · Ian Enting (Referee) · 2 Dec 2019

This is a valuable paper and the analysis is suitable for publication. However more careful wording is needed for a number of aspects.

This is a synthetic data study. That is an entirely reasonable thing to be doing. However, to say that the analysis is "using $CO_2$ observations from .. OCO-2" (as the authors do in the abstract and in the conclusion) is simply not true.

A complicating aspect is that the fluxes from carbon-tracker are themselves the product of an inverse calculation and so will have different spatio-temporal correlations than actual fluxes.

I would strongly disagree with the claims of uniqueness of GIM with regard to including

other types of information.

There are two different things:

1. The geostatistical approach of using spatio-temporal correlation structure as a technique for regularising an ill-conditioned inverse problem; and

2. the inclusion of additional information about fluxes (incorporated into GIM through $\beta$).

There is nothing to prevent the inclusion of additional information into inversion techniques that do not use geostatistical constraints.

While the specific form of $p(\mathbf{S}|\beta)$ that is used leads to linear equations and a direct solution, once direct solutions are replaced by 'variational' approaches, more general forms of $p(\mathbf{S}|\beta)$ can be incorporated, either with or without the use of regularisation by imposing a spatio-temporal correlation structure.

**page 11, Line 5**. The Miller et al. (2018) study doesn't seem to provide much information about the actual spatio-temporal correlation structure of the OCO-2 data (i.e. the structure of **R**. More discussion of this would be desirable.

Ian Enting, December 2019.

---

## Referee Comment (RC2) · Peter Rayner (Referee) · 15 Dec 2019

This paper describes some strategies for solving geostatistical inverse problems in tracer sources and sinks. As the authors note, these approaches form an important alternative to the more common classical Bayesian approach and have been widely and successfully used. These methods have thus far not often been scaled to deal with the large data sets and control vectors becoming usual in tracer inverse studies now. This paper describes some strategies for doing this. It is thus certainly within scope for GMD.

The paper is well-written and sound and makes a useful contribution for those wishing to follow such methods but held back by computational aspects. I have one comment

about such papers in general and two requests for the authors.

My first question concerns the code. This is likely to be a significant part of the contribution of the paper, at least for those who use MATLAB. Yet most people are not solving exactly the same problem as the authors. So the question arises how to make such code more generally useful, and from the journal's viewpoint, how to have its utility reviewed. I wonder if a short appendix to the paper or a document attached to the code describing any particular problems the authors had to overcome to implement the method and the approaches they took might be more generally useful than learning this from the code directly.

My other question concerns section 5.2. The general finding here is that the reduced rank approximation will overestimate posterior uncertainty since it reduces the size of the update made via the Sherman\-Morrison\-Woodbury matrix lemma. I agree with that but doesn't it also reduce the generalised variance of the prior by, for example, limiting the number of eigen-values in the decomposition? If that is correct do we have any sense of how this balance plays out?

Beyond these questions (neither of which need much work I think) I recommend the paper for publication.

---

## Author Comment (AC1) · 26 Dec 2019

We would first like to thank Ian Enting for his thoughtful review. His input and suggestions have been extremely helpful, and the manuscript is better for it. Below, we have listed Dr. Enting's comments in bold and discuss the associated changes we have made to the manuscript.

- **This is a synthetic data study. That is an entirely reasonable thing to be doing. However, to say that the analysis is "using CO2 observations from .. OCO-2" (as the authors do in the abstract and in the conclusion) is simply not true.**

  Dr. Enting makes a good point here. We do use synthetic observations, not real

observations. We have modified the text in the abstract to clearly indicate that these are synthetic observations.

- **A complicating aspect is that the fluxes from carbon-tracker are themselves the product of an inverse calculation and so will have different spatio-temporal correlations than actual fluxes.**

This is also a fair point. We do not know what actual $CO_2$ fluxes are, but a flux product like CarbonTracker is likely the closest approximation to actual fluxes that one can find. To this end, we have added a caveat on pg. 11, line 25 (of the original GMDD manuscript); real-world $CO_2$ fluxes may be different from the CarbonTracker fluxes used here. Hence, the analysis presented in this study is an approximation or prototypical example of the computational challenges one might encounter in an inverse model.

- **I would strongly disagree with the claims of uniqueness of GIM with regard to including other types of information.**

  **There are two different things:**

  1. **The geostatistical approach of using spatio-temporal correlation structure as a technique for regularising an ill-conditioned inverse problem; and**
  2. **The inclusion of additional information about fluxes (incorporated into GIM through $\beta$ ).**

  **There is nothing to prevent the inclusion of additional information into inversion techniques that do not use geostatistical constraints.**

  **While the specific form of $p(s|\beta)$ that is used leads to linear equations and a direct solution, once direct solutions are replaced by 'variational' approaches, more general forms of $p(s|\beta)$ can be incorporated, either with or**

**without the use of regularisation by imposing a spatio-temporal correlation structure.**

We agree with point #1. Recently, numerous inverse modeling studies that use a classical Bayesian approach have included spatio-temporal correlation structure in the prior covariance matrix (referred to in this manuscript as $\mathbf{Q}$). Hence, while some previous GIM studies have argued this is a unique feature of a GIM, we do not make that argument in the present manuscript. We have modified line 1 on page 4 to make this distinction clearer.

With regard to point #2: we agree that more general forms of $p(\boldsymbol{s}|\boldsymbol{\beta})$ could be incorporated, but this has not been done in existing atmospheric inverse modeling studies. I.e., there is no atmospheric inverse modeling study that we are aware of that has done that to date. In theory, one could design an inverse model in any number of ways with different distributional assumptions, prior probability densities, hyperparameters, and hyperpriors. However, nearly all atmospheric inverse modeling studies to date use a relatively narrow set of cost functions that have a similar form. In that context, the way in which GIMs have been applied to atmospheric inverse problems and the way in which these studies have formulated the prior probability is unique relative to what has been done to date in classical Bayesian inverse modeling studies using atmospheric trace gas observations.

- **page 11, Line 5. The Miller et al. (2018) study doesn't seem to provide much information about the actual spatio-temporal correlation structure of the OCO-2 data (i.e. the structure of $R$. More discussion of this would be desirable.**

We have added several sentences to the revised manuscript to clarify the inverse modeling approach used in the manuscript.

The actual error correlation length and correlation time varies depending upon the region in question and the day of the year, depending upon factors like ob-

servation type (e.g., nadir versus glint), atmospheric aerosol concentrations, and variations in surface albedo (e.g., O'Dell et al. 2018). In Miller et al. (2018, 2019), we estimated a spatial error correlation lengthscale between 0km and 3,800km for land nadir and land glint observations (version 9) using a spherical covariance model. We also calculated error correlation times between 0 days and 45 days, also using a spherical model.

In the present manuscript, we used a diagonal structure for $\mathbf{R}$. All existing inverse modeling studies that we are aware of using OCO-2 observations utilize a diagonal structure for $\mathbf{R}$, and we use a similar approach that is prototypcal of existing inverse modeling studies. Also note that the STILT footprints here were run every 2 seconds (approximately every 15km) along the OCO-2 flight track, so the synthetic observations used in this study are spaced further apart than the observations in the full OCO-2 dataset. In some locations, this spacing is comparable to the estimated spatial error correlation length in the OCO-2 observations.

It may be helpful to incorporate off-diagonal elements within $\mathbf{R}$ for future inverse modeling studies – to better account for the information content of the OCO-2 observations and/or to estimate rigorous posterior uncertainties. However, that has not been done in existing OCO-2 studies to date, and doing so would likely necessitate accounting for the highly non-stationary structure of these off-diagonal elements. We discuss the possibility of incorporating off-diagonal elements within $\mathbf{R}$ in the manuscript (e.g., pg. 13 of the GMDD manuscript). The overall focus of this paper is on computational approaches to large inverse problems, and to that end, we felt that developing an approach to describe non-stationary error covariances in $\mathbf{R}$ was beyond the scope of the current manuscript.

**References**

Miller, S. M., Michalak, A. M., Yadav, V., and Tadić, J. M.: Characterizing biospheric carbon balance using $CO_2$ observations from the OCO-2 satellite, Atmos. Chem. Phys.,

18, 6785–6799, https://doi.org/10.5194/acp-18-6785-2018, 2018.

Miller, S. M., Michalak, A. M.: The impact of improved satellite retrievals on estimates of biospheric carbon balance, Atmos. Chem. Phys. Discuss., https://doi.org/10.5194/acp-2019-382, in review, 2019.

O'Dell, C. W., Eldering, A., Wennberg, P. O., Crisp, D., Gunson, M. R., Fisher, B., Frankenberg, C., Kiel, M., Lindqvist, H., Mandrake, L., Merrelli, A., Natraj, V., Nelson, R. R., Osterman, G. B., Payne, V. H., Taylor, T. E., Wunch, D., Drouin, B. J., Oyafuso, F., Chang, A., McDuffie, J., Smyth, M., Baker, D. F., Basu, S., Chevallier, F., Crowell, S. M. R., Feng, L., Palmer, P. I., Dubey, M., García, O. E., Griffith, D. W. T., Hase, F., Iraci, L. T., Kivi, R., Morino, I., Notholt, J., Ohyama, H., Petri, C., Roehl, C. M., Sha, M. K., Strong, K., Sussmann, R., Te, Y., Uchino, O., and Velazco, V. A.: Improved retrievals of carbon dioxide from Orbiting Carbon Observatory-2 with the version 8 ACOS algorithm, Atmos. Meas. Tech., 11, 6539–6576, 10.5194/amt-11-6539-2018, 2018.
* * *

---

## Author Comment (AC2) · 26 Dec 2019

We would like to thank Peter Rayner for his feedback and constructive ideas on the manuscript. These ideas have been very helpful for improving the overall quality of the manuscript and strength of the analysis. We have listed Dr. Rayner's comments in bold typeface below and discuss the associated changes we have made to the manuscript.

- **My first question concerns the code. This is likely to be a significant part of the contribution of the paper, at least for those who use MATLAB. Yet most people are not solving exactly the same problem as the authors. So the question arises how to make such code more generally useful, and from the journal's viewpoint, how to have its utility reviewed. I wonder if a short**

[Figure]

**appendix to the paper or a document attached to the code describing any particular problems the authors had to overcome to implement the method and the approaches they took might be more generally useful than learning this from the code directly.**

We think this is a great suggestion and have added additional text to the user guide for the associated model code on Github and Zenodo. Much of this text is specific to the practicalities of coding the concepts described in the paper, so we have included this text with the code instead of as an appendix to the manuscript. Below, we have pasted the additional text that has been added to the user guide.

*Additional text that will be added to the software user guide:*

The computational approaches implemented in this code are designed for large inverse problems, but it is nevertheless important to keep computational considerations in mind when adapting the code for a specific inverse problem. We discuss several of these considerations below:

1. The number of iterations required by the iterative solver to estimate the fluxes can be an important limiting factor when using certain types of adjoint atmospheric models but may not be a limiting factor when using other types of atmospheric models. For trajectory models like the Stochastic Time-Inverted Lagrangian Transport (STILT) model, $\mathbf{H}$ is formulated explicitly and can be read in directly. In that case, the computing resources required to run numerous STILT trajectories, not the number of iterations required by the solver, is likely to be the computational bottleneck. By contrast, the number of iterations required to converge on a solution is likely to be the bottleneck for gridded chemical transport models like GEOS-Chem or TM5. These models do not produce an explicit $\mathbf{H}$ and $\mathbf{H}^T$ matrices, and one must instead run the forward and adjoint models once per iteration of the solver. These calculations often become time-intensive when numerous iterations are required to converge on a flux estimate. Furthermore, some adjoint

models (i.e., GEOS-Chem) cannot be run in parallel for greenhouse gas applications, though we expect that these capabilities will change in the future with the development of an adjoint for models like GEOS-Chem-High Performance (GC-HP).

2. The matrices $\mathbf{D}$ and $\mathbf{E}$ (Eqs. 9-10 in the manuscript) are usually straightforward to store in memory and/or invert given the dimensions of most atmospheric inverse models to date. However we anticipate that this will change in the future as atmospheric models like GEOS-Chem have better parallel computing capabilities and can be run at higher spatial resolution. In those cases, it may be important to structure $\mathbf{D}$ and $\mathbf{E}$ as hierarchical matrices or circulant matrices to avoid problems with storing these matrices in memory or inverting these matrices.

3. The choice of covariance function can have a large impact on the wall clock time and memory required for matrix calculations using $\mathbf{D}$ and $\mathbf{E}$. An exponential covariance model is very common in existing GIM studies in hydrology and atmospheric science. For large inverse problems, this choice may not be practical; an exponential model will never decay to zero. As a result, $\mathbf{D}$ and $\mathbf{E}$ will never be sparse matrices. By contrast, other covariance models, like a spherical model, do decay to zero, and $\mathbf{D}$ and $\mathbf{E}$ can be formulated as memory-saving sparse matrices.

4. The code here can be re-written for other languages if a different language is more convenient than Matlab. We recommend that users exercise caution if doing so because different commonly-used languages can exhibit very different performance. For example, we found that R is far slower than Matlab at linear algebra and often requires more memory than Matlab for the same matrix inversion.

- **My other question concerns section 5.2. The general finding here is that the reduced rank approximation will overestimate posterior uncertainty since**

**it reduces the size of the update made via the ShermanMorrisonWoodbury matrix lemma. I agree with that but doesn't it also reduce the generalised variance of the prior by, for example, limiting the number of eigen-values in the decomposition? If that is correct do we have any sense of how this balance plays out?**

We have clarified this point in the text. In this setup, the prior is taken to be a positive definite, full rank matrix and is not affected at all by the approximation; however, the posterior covariance is written as an update of the prior covariance matrix involving selected eigenpairs. Since we are subtracting a positive semidefinite update, this ensures that the variance is reduced. An intuitive way of understanding is that by observing data, the variance (i.e., the uncertainty) is reduced since we know more about the parameters of interest.